

# The use of ground-based GNSS for atmospheric water vapour variation study in Papua New Guinea and its response to ENSO events

Ansaldi Senat[1], Yidong Lou[2], Weixing Zhang[2], Jingna Bai[2], Chuang Shi[3]

[1]State Key Laboratory of Information Engineering in Surveying Mapping and Remote Sensing (LIESMARS), Wuhan University, Wuhan, China
[2]Global Navigation Satellite Systems Research Center, Wuhan University, Wuhan, China
[3]School of Electronic and Information Engineering, Beishan University, Beijing, China

*Correspondence to*: Weixing Zhang (zhangweixing89@whu.edu.cn)

**Abstract.** The spatial and temporal variability distribution of atmospheric water vapour in Papua New Guinea region is investigated using three ground-based GNSS station datasets and are compared with radiosonde data and the ERA-Interim reanalysis to generate the atmospheric precipitable water vapour (PWV) products over PNG from 2000 to 2019. From this product, PWV variations on multiple timescales are studied, with the water vapour products of GNSS and ERA-Interim in good agreement with their large-scale changes, which is reflective of the large-scale water vapour transport. At daily periods, the diurnal amplitudes of GNSS is larger at the mainland station (3.5mm) than the two island stations (1-1.8mm). The ERA-Interim amplitudes are smaller than GNSS on a daily basis, and do not capture the diurnal phases correctly. The estimated long-term PWV linear trends are predominantly positive and statistically significant which is in agreement in sign to the increase in moisture expected by the Clausius-Clapeyron equation under the background of global temperature rise. In addition, the regional impact of PWV in PNG in response to the El Niño- Southern Oscillation events are analysed using a correlation analysis, focusing on the dynamic influence of the large-scale nature of the 2010-2012 Bimodal La Niña and 2015-2016 El Niño events. The sea surface temperature anomaly in the Niño 3.4 and Niño 4 regions are selected to describe these two events. Both events portray overall negative correlation characteristics at the three GNSS stations with stations PNGM and RVO_ showing the strongest correlation during the 2010-2011 La Niña event significant at a 99% confidence level.

## 1 Introduction

Water vapour is the principal greenhouse gas component in the atmosphere showing significant variability in its distribution; with seasonal and regional variations as a result from interactions between atmospheric and surface processes (Bock et al., 2007). This performs a key role in the global energy budget through its radiative characteristics and thermodynamic properties (Kiehl and Trenberth, 1997; Mengistu et al., 2015), and is an important climate variable in extreme weather events (Ssenyunzi et al., 2020). From a climatic standpoint, knowing about the accurate distribution and variability of water vapour in its global





content is essential to climate studies and weather forecasting, and could potentially be used as an indicator for understanding, modelling and predicting the global weather and climate processes (Yuan et al., 1993; Vey et al., 2010).

The measurement of water vapour still remains one of the least atmospheric parameters observed, both temporally and spatially. The traditional standard method of radiosondes (Durre et al., 2006), and satellite observation techniques of passive infrared sounders (Rocken et al., 1995) and microwave radiometers (Liou et al., 2001) have previously been used in many

studies to estimate the variability of water vapour. However, the limitations in these techniques by issues of calibration, poor quality of data, temporal resolution affect the accuracy of water vapour measurements and its long-term reliability (Choy et al., 2015).

In recent years, ground-based GNSS measurements have proven to accurately estimate water vapour with high temporal and spatial resolution, with the major advantage of having homogeneity over the long term (Bevis et al., 1992). Water vapour

measurement may be expressed in terms of Precipitable Water Vapour (PWV) which is the amount of water (in millimetres) that would result from condensing a column of water that extends from the measurement point to altitudes of about 12 km (Alshawaf et al., 2017), by retrieval from propagation delays in the GNSS signals through the atmosphere and collected at ground-based permanent GNSS receivers at similar levels of accuracy to radiosondes and radiometers (Elgered et al., 1997; Emardson et al., 1998). Ground-based GNSS therefore can serve as an independent validation dataset for quantifying or

calibrating errors in radiosonde observations (Wang and Zhang, 2008) and assimilation into numerical weather prediction (NWP) and climate models to improve prediction system accuracies (Baker et al., 2001; Vedel and Huang, 2004).  With this regard, this study uses the ERA- Interim derived PWV released by the European Centre for Medium-Range Weather Forecasts (ECMWF) as a comparative data set to assess the performance of GNSS PWV, to investigate the water vapour cycle properties at seasonal and diurnal timescales and to evaluate the capacity of ERA-Interim to capture the spatial and temporal distribution

of water vapour over the Papua New Guinea (PNG) region.

Despite the establishment of many GNSS ground stations over the tropical south pacific region in recent decades, PWV estimation still is a challenge particularly over the PNG region.  This is due to a scarce coverage of surface meteorological stations over PNG with large data gaps. Limitation in accurate and reliable meteorological measurements affect the accuracy of GNSS PWV estimates (Vey et al., 2009). Data from NWP models like ECMWF or, other empirical meteorological models

like the Global Pressure and Temperature (GPT) (Boehm et al., 2007) are applied if surface meteorological measurements are not available (Chen et al., 2018). The model accuracies shall vary and are dependent on the model, region, season and other climate conditions (Ssenyunzi et al., 2020). Therefore, this study uses $T_m$ and surface pressure derived from ERA-Interim to compute PWV.

Considerable efforts in many studies have focused on climate monitoring by quantifying and analysing the trends in PWV

time series from radiosonde (Ross and Elliot, 2001; Durre et al., 2009; Wang et al., 2016; Makama and Lim, 2020), ground-based GNSS (Gradinarsky et al., 2002; Zhang et al., 2017) and reanalysis products and model data (Trenberth et al., 2005; Wagner et al., 2006; Dessler and Davis, 2010; Zhang et al., 2013) in different regions and time periods. Identifying the statistical significance of PWV estimated trends is important because detecting the correct trend depends on its magnitude, the



autocorrelation in the time series, and the length (Alshawaf et al., 2018). Radiosondes and microwave radiometer (MWR) have
been used to estimate PWV trends over the South Pacific region including PNG for a long period of time, for example from
1988- 2011 in Wang et al. (2016). Chen and Liu (2016) used datasets from ERA-Interim and NCEP reanalysis to investigate
the global PWV trends from 1979 to 2014, and determined a positive trend of 0.01 ±0.15 mm decade$^{-1}$ and 0.13 ±0.15 mm
decade$^{-1}$ for the tropical region from both reanalysis products. However, reanalysis data may be susceptible to time-varying
biases indicating that the derived trend may not by reliable (Sherwood et al., 2010). This may be linked to the input of
radiosonde data, for unadjusted inhomogeneity and biases from radiosonde observations assimilated into published reanalysis
products will incite erroneous changes in the long-term water vapour trends (Elliot and Gaffen, 1991; Easterling and Peterson,
1995).

At present, ground-based GNSS time series are relatively long and can be used to estimate climatic trends. However, with
the time period shorter than the 30-year climate normal defined by climate scientists to exclude the interannual variations or
anomalies in weather and evaluate climate effects for a particular site as described by the World Meteorological Organization
(WMO) (Arguez and Vose, 2011), this is merely time dependent. The long-term stability of the time series is also another key
issue in climate applications (Pacione et al., 2017). This may be affected by changes in GNSS processing, causing
inconsistencies of several millimetres in GNSS PWV (Steigenberger et al., 2007). Nevertheless, these studies motivate this
study to analyse the long-term trends in PWV over the PNG region.

In addition to this study, the response of GNSS PWV to ENSO events is also investigated. ENSO portrays a dominant
influence on the global climate system at interannual timescales (Wagner et al., 2005), however the influence of ENSO on the
climate is not uniform and is dependent on the intensity of each ENSO event, with respect to the time of the year it develops
and its interaction with other key climate drivers and patterns (Wang et al., 2018). This makes it challenging for climate
scientists to predict their occurrences. ENSO has been found to modulate the interannual variation of the water vapour
component of the atmosphere, mainly through the convective transport of lower tropospheric water vapour (Su and Jiang,
2013; Tian et al., 2018). As ENSO is a coupled ocean-atmosphere interaction phenomenon with relations to sea surface
temperature (SSTs) and easterly trade winds in the tropical Pacific Ocean, atmospheric water vapour can be used to trace
ENSO evolutions (Wallace et al., 1998; Seidel, 2002; Wagner et al., 2005; Wang et al., 2018). In addition, the strong correlation
between PWV and precipitation can also serve as a measurement to understand the impacts of ENSO events (Trenberth et al.,
90   2005).

Studies covering the influence of ENSO on water vapour in the atmosphere focus mainly on the tropical regions (Ferraro
et al., 1996; Soden, 2000; Foster et al., 2000; Lu et al., 2004; Wagner et al., 2005). Several of these studies investigate this
relation using microwave radiometer observations due to the lack of radiosonde stations over the tropics (Vey et al., 2009).
However, the reliability of PWV estimates from satellite microwave radiometers is acceptable only over the oceans (Wentz,
1997; Allan et al., 2004), which can be complemented on land by use of GNSS observation. Foster et al. (2000) demonstrated
the impact of 1997/1998 El Niño on GPS-PWV by evaluating two GPS stations in the tropical western Pacific between the
time period from 1997 to 1998. Suparta et al. (2017) analysed the dynamics of ENSO events (La Niña and El Niño) in the





western tropical pacific during 2009-2011 using GPS-PWV. Similar to the latter study, this research investigates the response of GNSS PWV to two major ENSO events that impacted PNG in the last decade.

The principle aim of this study is to provide knowledge of the distribution and variability of atmospheric water vapour over PNG using ground-based GNSS, radiosonde and ERA-Interim reanalysis, which is crucial to understanding the hydrological processes which plays a pivotal role in climate regulation.

## 2 Data and Methodology

### 2.1 Study Area

The island country of PNG occupies the eastern half of the island of New Guinea. It is located at the junction of the equatorial Indo-Malayan and South West Pacific regions and is distinctively referred to as part of the 'maritime continent' (Ramage, 1968). Situated close to the equator (between 2 and 12°S), PNG lies within the Western Pacific Warm Pool (WPWP) region. This tropical ocean feature appears as the principal source of water vapour and for regulating global heat transfer, with sea surface temperatures (SST) reaching a maximum of ~30° (Pierrehumbert, 2000).


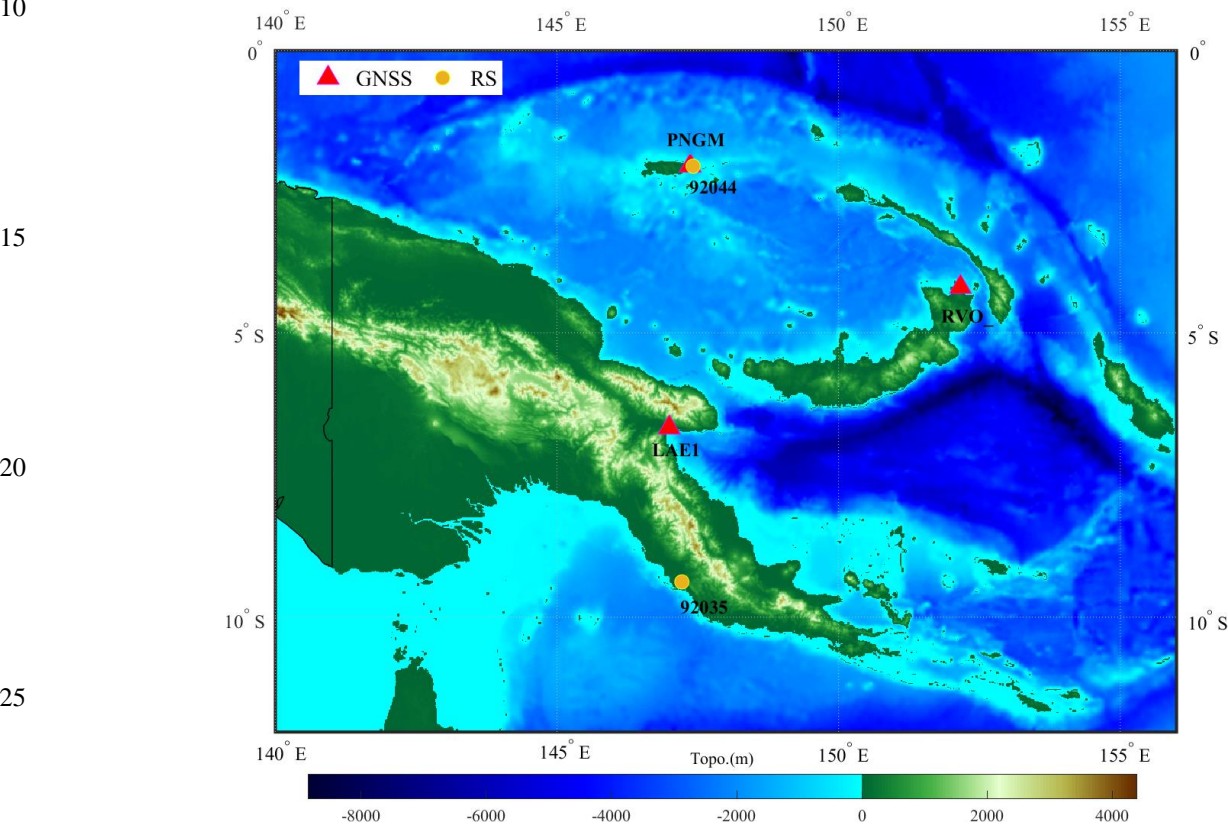

**Figure 1.** Geographical location of the three GNSS stations (red triangle) and two radiosonde sites (yellow circle) over PNG.





The country has a tropical climate and experiences some of the highest rainfall totals on earth, with mean annual rainfall totals ranging from about 1000mm to over 8000mm, with the largest portion of PNG receiving between 2000mm to 4000mm per annum (McAlphine et al., 1983). Most parts of PNG experience rainfall patterns exhibiting some seasonal distribution with the maxima occurring between December and April during the 'north-west' season and the minima during the 'south-east' season from May to October, influenced by the discontinuous and moving Intertropical Convergence Zone (ITCZ) and the

regular alternation between the two major airstreams of the southeast trade winds and the north-western monsoon (Hall, 1984). Global climate drivers such as the ENSO and the Madden-Julian Oscillation (MJO) (Madden and Julian, 1994) also impose large scale variations on these seasonal rainfalls. In addition, mesoscale climates are also produced by the interactions between the regional airflow and the country's physiography. This drives smaller-scale variations (i.e. land-sea breezes, mountain-valley winds; mesoscale convective complexes) which influences local rainfall variability (Robbins, 2016).

The study will focus primarily on the three wet coastal lowland (>200m) areas of Morobe, Manus and the East New Britain provinces of PNG, where the GNSS stations are located respective to the two radiosonde sites at Manus Province and the National Capital District (Fig. 1). These wetter lowland areas have a range of mean annual rainfall values that is much greater compared to the central highland areas of the mainland (Bellamy, 1995). These areas have the greatest percentage of annual rain falling at high intensities, and also accounts for the great amounts of water vapour being continuously evaporated from

both the surrounding tropical ocean and the land itself (McAlpine et al., 1983). Therefore, knowing the importance and distribution of water vapour over PNG is crucial to a better understanding of the climate and climate change (Starr and Melfi, 1991).

### 2.2 GNSS PWV Products

**Table 1.** The observation periods of each GNSS and Radiosonde sites covered in the study.

| Location | Site | Network | GNSS Period | ERA-Interim | IGRA site | IGRA Period |
|---|---|---|---|---|---|---|
| Lae, Morobe Province | LAE1 | IGS & IGS Reference Frame site | Jan 2001- Dec 2019 | 2000-2019 | - | - |
| Manus, Manus Province | PNGM | PSLM | May 2002 - Dec 2019 | 2000-2019 | 92044 | 2002-2013 |
| Rabaul, East New Britain | RVO_ | Rabaul Caldera Network | May 2000 - Oct 2018 | 2000-2019 | - | - |
| Port Moresby, NCD | - | - | - | - | 92035 | 2000-2013 |





Three GNSS stations in the PNG region (Table 1) are used to generate 6-hourly GNSS PWV products, covering the period between 2000-2019. The details of GNSS data processing and PWV retrieval are referenced to Zhang et al. (2017), but only

described briefly in this section.

The PANDA (position and navigation data analyst) software package (Shi et al., 2008) was used to process daily GNSS measurements of the three stations, by the precise point positioning (PPP) method. Each daily GNSS measurements were sampled at 30 second intervals. The second reprocessing (repro2) final satellite orbit and clock products provided by the IGS data analysis center, the European Space Agency (ESA), between 2000 to early 2014 i.e. ES2, and the original ESA final

products from early 2014 to 2019 were used. These final products used are reprocessed in the IGb08 reference frame and have been fixed in the PPP method. Differential code biases between frequencies from the Center for Orbit Determination in Europe (CODE) were employed (Schaer and Steigenberger, 2006). Correction models such as the absolute antenna phase center correction model (Schmid et al., 2007), phase wind-up corrections (Wu et al., 1993), and the relativity corrections were applied. The ground station coordinates are estimated as diurnal constants and receiver clocks are solved at every epoch with errors

taken as white noise.

The zenith total delay is the estimation of the total propagation delay and can be divided into two components referred to as the zenith hydrological delay (ZHD) and the non-hydrological delay, also called the zenith wet delay (ZWD) (Barreto et al., 2013; Ansari et al., 2018). Based on the Saastamoinen model (Saastamoinen, 1972), the a priori ZHD and a priori ZWD were estimated using the empirical global pressure and temperature (GPT) model (Boehm et al., 2007) and 50% relative humidity

at mean sea level (MSL). Estimated corrections to the a priori ZWD are derived as piecewise constant every 2 hours with a power density of 20 mm/$\sqrt{h}$ , with $h$ denoting the time in hours. The a prior ZHD, the a priori ZWD, and the estimated ZWD corrections are summed up to obtain the final ZTD. The global mapping function (GMF) (Boehm et al., 2006) was used and the horizontal tropospheric gradients in the north-south and east-west directions were estimated at 12-hour interval (Lou et al., 2018). The cut off elevation of 7° was set and an elevation dependent weighting strategy was applied to low elevation

observations (below 30°) (Gendt et al., 2003).

The air pressures at GNSS stations were acquired from ERA- Interim products and used to calculate precise values for ZHD. ERA-Interim products were used because all stations are not equipped with meteorological sensors or have no nearby weather observation station records. ERA-Interim derived pressures are used to calculate precise ZHD values, which is subtracted from ZTD to acquire ZWD. The weighted temperature $T_m$ estimated from ERA-Interim are then used to convert

ZWD values to PWV. The accuracy of the Π coefficient and the ZWD are influenced by ERA-Interim derived pressure and mean temperature (Ning et al., 2016).

## 2.3 Radiosonde data

Radiosonde observations for the two IGRA (Integrated Global Radiosonde Archive) stations 92044 and 92035 in PNG are derived using the GNSS Meteorological Ensemble Tools (GMET) online service (http://gmet.users.sgg.whu.edu.cn/)

supported and maintained by the GNSS Research Centre, Wuhan University. This website estimates tropospheric parameters



including PWV for global IGRA stations. Two radiosonde stations are observed between the years 2000 to their last launch years in 2013. The radiosondes are usually launched two times a day (usually at around 0000 and 1200 UTC) where the IGRA archive contains quality-assured data (Durre et al., 2006). However, these datasets contain sounding data with missing values as it does not account for solar radiation bias correction or interpolated data gaps. Furthermore, most measurements have

insufficient altitude coverage to calculate PWV. Therefore, limited radiosonde observations with altitude data exceeding 200 hPa are available for the following intercomparison (Mengistu et al., 2015).

As radiosonde launch sites and GNSS stations are usually not collocated, this study will compare only one matched pair on Manus island during the 2002-2013 period. The radiosonde-GNSS station site location difference satisfies the requirements described by Zhang et al. (2017). The radiosonde station 92044 at Momote airport on Manus island undergoes an IGRA quality

assurance system check based on scrutiny for the presence of physical implausible values, internal inconsistencies among variables, climatological outliers, and temporal and vertical inconsistencies in temperature (Durre et al., 2006). The radiosonde type used at Momote is the Vaisala RS80-H (Long, 2014). The Vaisala RS80H-type "humicap" sensor is known to suffer from dry (negative) bias in relative humidity by several percent, due to its more sensitivity to water vapour and, more expectedly, more contamination as well (Miller et al., 1999; Nakamura et al., 2004).

**2.4 ERA-Interim**

ERA-Interim is the third-generation global atmospheric reanalysis data produced by the ECMWF covering the time period from 1979 to August 2019 (Dee et al., 2011). The dataset contains a spatial resolution of approximately 80 km (0.75° x 0.75°) on 60 model levels from the surface up to 0.1 hPa (an altitude of about 65 km) (Gregow et al., 2015). Although the assimilation of radiosonde observations serves as a fundamental data source in the reanalysis, the analysis results of humidity in the tropical

regions is of a much poorer quality due to the lower number of accurate radiosonde observations available (Wang et al., 2002b; Vey et al., 2010). The accuracy of PWV from ERA-Interim over Papua New Guinea will be quantitatively assessed by comparison with GNSS data between the period from 2000 to 2019, as ERA-Interim does not incorporate the assimilation of GNSS dataset.

Utilizing a 4-dimensional variational analysis (4D-Var) with a 12-hour analysis window, the ERA-Interim provides a

temporal resolution at 00:00, 06:00, 12:00, and 18:00UTC. The daily fields (surface pressure, temperature and specific humidity) on pressure levels are used (data available online at https://apps.ecmwf.int/datasets/data/interim-full-daily ). Each GNSS stations are surrounded by four grid boxes with individual field values that are interpolated horizontally to the GNSS station positions at each pressure level (Jade and Vijayan, 2008). The altitude offset correction method is then applied to correct the variables at pressure levels obtained at the horizontal locations of the GNSS antenna to calculate PWV from the GNSS

antenna level to the top level of ERA-Interim, as the as the topographical model using in ERA-Interim does not coincide with the GNSS station height (Berckmans et al., 2018).





## 2.5 ENSO SSTA data

Figure 2 shows the variation of the Niño 3.4 (5N-5S, 120W- 170W) and Niño 4 (5N- 5S, 150W- 160E) weekly Optimum Interpolation Sea Surface Temperature version 2 (OISST.v2) indices between 2010 to 2017, which were measures to quantify

the intensities of the occurrence of the natural interannual climate variability phenomena of ENSO, with the studied regions highlighted in blue and red. The weekly OISST v2 are computed weekly by using a statistical method to combine different forms of observations (satellite, ships, buoy, Agro floats) and fitting them onto a 1° latitude by 1° longitude gridded version of the world (Reynolds et al., 2002). Data are quality controlled with improved ship and buoy track checks to remove observations with bad locations (Reynolds, 1988). OISST were obtained from NOAA and used to observe the small-scale

spatial and more frequent temporal changes of SST within the Niño 3.4 and Niño 4 regions during two major ENSO events that affected PNG in the last decade; i.e. the 2010 – 2012 bimodal La Niña and the 2015 – 2016 El Niño events.

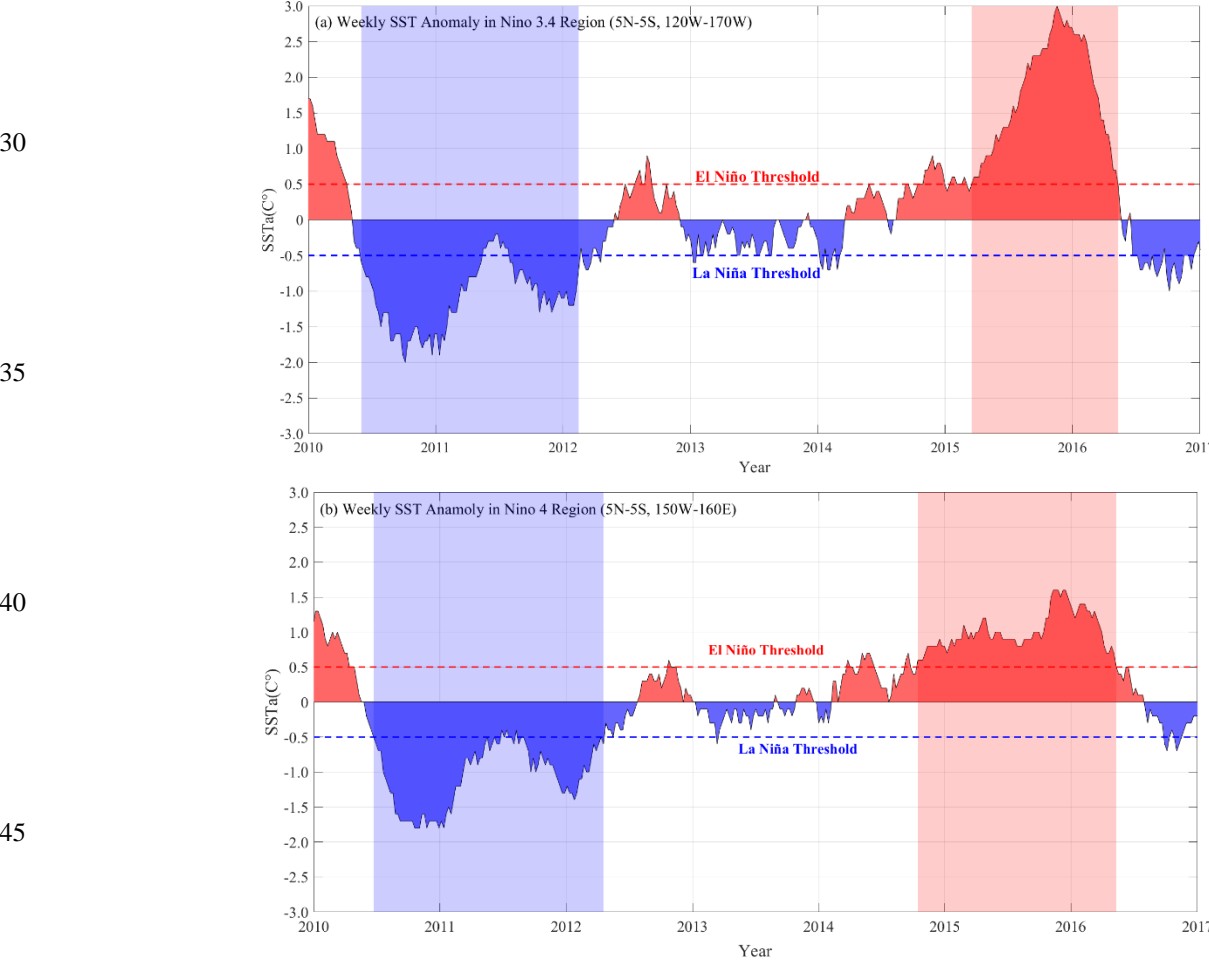

**Figure 2.** The weekly SSTa variations for the (a) Niño 3.4 region and (b) Niño 4 region. The highlighted regions of the two plots are the occurrences of the two ENSO events analysed in the study. Blue highlighted region indicates the 2010- 2012 Bimodal La Niña events and the red highlighted region is the 2015 – 2016 El Niño event.






The relationship between GNSS PWV and weekly SSTa variability is studied using a correlation analysis. To observe the behaviour of PWV variability during ENSO occurrences, two indications at identifying ENSO events are defined. The first is the SST anomalies of the Niño 3.4 region which is one of the two official NOAA ENSO indices (the other being the Oceanic Niño Index (ONI)). In this regard, ENSO events (El Niño/ La Niña) are declared to have occurred when the 5-month running

mean of Niño 3.4 SST anomalies exceeds +0.4°C (-0.4°C) for six months or more (Trenberth and Hoar, 1997).

The second definition by the Japan Meteorological Agency (JMA) declares an ENSO year (El Niño/ La Niña) when the JMA index values surpass ±0.5°C for six consecutive 5-month periods (Hong et al., 2001; Hanley et al., 2003; McGregor and Ebi, 2018). This approach calculates the ENSO index through a 5-month running mean of averaged spatial SST anomalies at the geographical range of 4°S - 4°N, 150° W- 90°W - latitude restricted Niño 3 region. Applying the 5-month running mean

of SSTa averages out the intra-seasonal variations in the tropical ocean, ensuring SSTa as a good indicator for ENSO activities. With these two definitions in place, it is proposed with confidence that an ENSO event will occur when SSTa >±0.5 °C and is used as the basis for investigating the relationship between GNSS PWV and SSTa.

## 3 Intercomparison of $T_m$ and PWV between the different instruments and ERA- Interim at Manus Island

Manus island is used as the case study to present the reliability of $T_m$ derived from ERA- Interim for GNSS PWV with
comparisons to IGRA site 92044 measurements and their consistencies with each other. IGRA site 92035 site in Port Moresby is studied for trend analysis only as no colocated GNSS station measurements are available for comparison within its time window. A statistical comparison is performed for the two daily radiosonde launch times at identically matched epochs, meaning that no interpolation was applied.

### 3.1 The $T_m$ error analysis

The accuracy of GNSS PWV is proportional to the accuracy of $T_m$, as it is an important parameter applied to retrieve PWV from ground based GNSS. As a function of atmospheric temperature and vertical humidity profiles, $T_m$ at GNSS sites can be well approximated based on a linear empirical relationship between surface temperature and $T_m$ (Bevis et al., 1992). However, this linear relationship noted an underestimation of $T_m$ by up to 6 K in the tropics and subtropics (Wang et al., 2005). For better accuracy, ERA-Interim reanalysis is used to derive $T_m$ at the GNSS stations. Manus site is the case study area to assess

the $T_m$ derived from the reanalysis with IGRA site 92044. Table 2 shows the comparisons performed for matching epochs between the period 2000-2013. The analysis at Manus found a mean bias (MnB) of -0.68 K and a root mean square (RMS) of 1.1 K respectively.




**Table 2.** Statistical comparison between ERA-Interim $T_m$ and radiosonde 92044 $T_m$

| Site | Period | ERA-Interim $T_m$ – IGRA site 92044 $T_m$ | | Number of counts |
|------|--------|------------------|---------|------------------|
| | | MnB (K) | RMS (K) | |
| PNGM | 2000-2013 | -0.684 | 1.053 | 5152 |

## 3.2 Comparison between GNSS PWV and Radiosonde PWV

IGRA site 92044 at Manus island is situated at Momote airport and is approximately 6 km from the GNSS station PNGM, with a height difference of 40.8m. The period from May 2002 (start of GNSS observations) to December 2013 (end of radiosonde observations) is analysed.

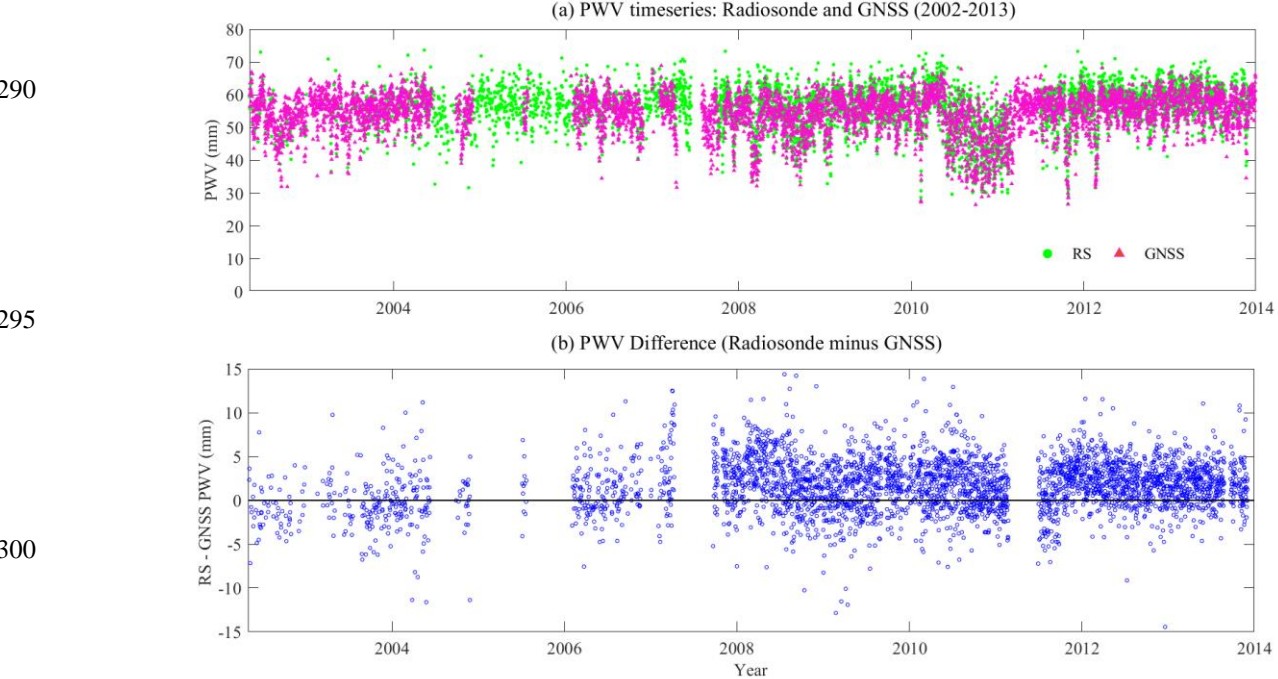

Figure 3. Timeseries of PWV between GNSS (magenta triangle) and radiosonde (green circle) from period 2002 to 2013 (top panel) and PWV differences (mm) between radiosonde and GNSS measurements over Manus (bottom panel). Positive values indicate that the
measurements are moister than GNSS measurements.

Figure 3 shows that the annual variability of PWV between the two instruments are quite consistent. Radiosondes are evaluated to having its own systematic bias depending on the types of radiosonde and should not be assumed to provide





certainty when assessing PWV intertechnique biases (Wang et al., 2007). Nevertheless, the traditional use of radiosonde
measurements of PWV to qualitatively assess GNSS derived PWV estimates is selected as the basis for comparison is the
study. A total of 3637 GNSS observations were paired and compared over the 11-year period.

**Table 3.** Approximate lateral distance (km), height difference (m), mean (mm), standard deviation (mm) and RMS (mm) of the differences
between RS and GNSS derived PWV estimates.

| RS – GNSS PWV | Period | GNSS-RS lateral (km) | GNSS-RS height (m) | Mean (mm) | RMS (mm) | Number of counts |
|---|---|---|---|---|---|---|
| 92044 – PNGM | 2002-2013 | 5.7 | 40.8 | 1.71 | 3.46 | 3637 |

Outliers were not discarded in the statistical analysis as they expect to exist in large datasets. This might contribute to the
slightly large RMS of PWV difference between radiosonde and GNSS, which is common. The mean values were calculated
based on the average differences between radiosonde and GNSS measurements of PWV, i.e. radiosonde minus GNSS. The
majority of radiosonde measurements of PWV are greater than GNSS estimates, and indicates the systematic bias between the
two instruments (Choy et al., 2015). Table 3 presents that the PWV mean difference between GNSS and radiosonde has a
moist (positive) bias of 1.71 mm and RMS difference of 3.46 mm respectively. One main cause of this large difference may
be the calibration uncertainties of radiosonde moisture sensors (Wu et al., 2003).

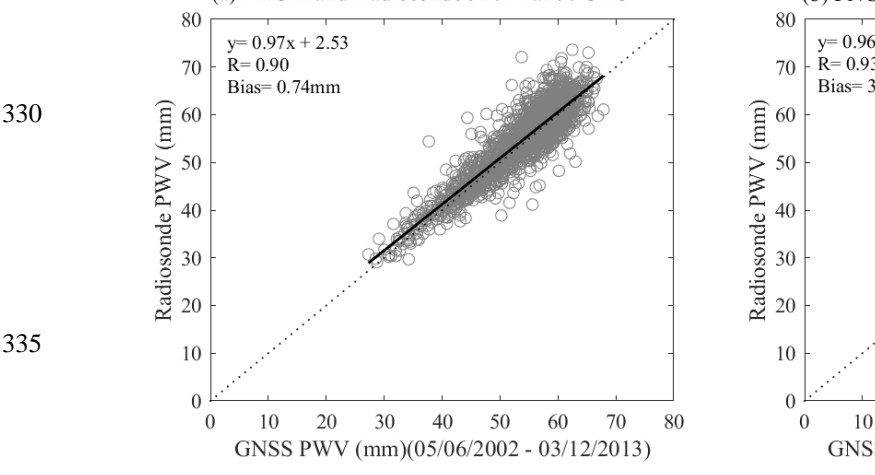
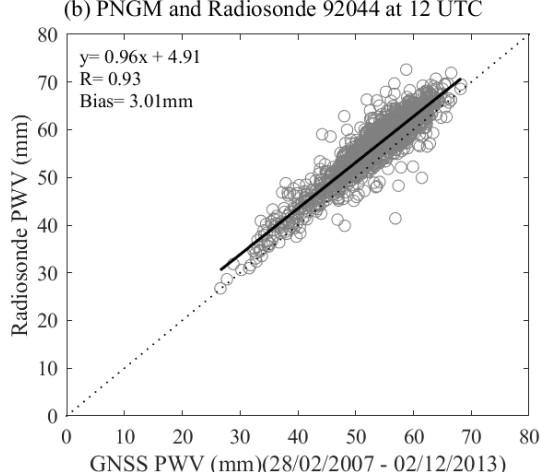

**Figure 4.** Scatterplots of GNSS vs. radiosonde PWV estimates at 0000 UTC and 1200 UTC.






Presented in Fig. 4(a) is the dry bias indicated by the increased difference between radiosonde and GNSS PWV for high values of PWV (>65mm). The overall wet bias at both times may be due to the overestimating of water vapour by radiosonde as moist biases are known to exist for large PWV ranges in Vaisala radiosonde types (Wang et al., 2002b). The standard deviation of 3 mm reflects the high atmospheric moisture content of the area.

Further inspection into the launch times of radiosonde in Fig. 4 revealed significant changes in the mean difference, with a low moist bias (0.74mm) at 00 UTC and a weak moist bias (3mm) at 12 UTC. This may be related to the type of capacitive humidity sensor in the Vaisala RS80 instrument reported by Wang and Zhang (2008). The large moist bias at day time is generally attributed to the solar radiation of the radiosonde humidity sensor (Turner et al., 2003). The slope of regression for the daytime and night time observations are 0.96 and 0.97, with a correlation of 0.93 and 0.90 respectively. The results

presented demonstrate that GNSS and radiosonde PWV measurements portray a generally good level of agreement, respective to the launch times.

**3.3 Radiosonde PWV Trends**

PWV linear trend estimation is susceptible to systematic effects in the time series caused by operational changes of GNSS or radiosonde sites (Campbell, 2003). Analysing the time series of a pre-set period is of importance to have an insightful and

reasonable conclusion about the long-term variations of PWV. However, because the PWV trends of the two IGRA sites is estimated based on different time windows within the research period at 7 to 13 years (between 2000 to 2013), these time series are analysed with respect to their launch times (0000 UTC and 1200 UTC).

The PWV trends are estimated using Sen's nonparametric method, which is often adopted by the meteorological community for trend estimations. The method gives a more robust slope estimation of a data set despite containing outliers or

extreme values (Fan and Yao, 2003). The PWV monthly anomalies are firstly computed as the deviations of the monthly PWV climatology (mean value of monthly PWV over the observed period – from 2000 to 2013) for radiosondes. The use of a 3-point moving average method is performed to the monthly PWV anomaly time series, and the PWV trends are estimated as:

$$\text{trend} = \text{median} \left( \frac{x_j - x_i}{j - i} \right) \tag{1}$$


With $x_j$ and $x_i$ representing the values of the monthly PWV anomaly at time $j$ and $i$ ($j > i$), respectively. The Mann Kendall tau method is also applied to test the statistical significance of the trends (Mann, 1945; Kendall, 1975) with 95% confidence level.




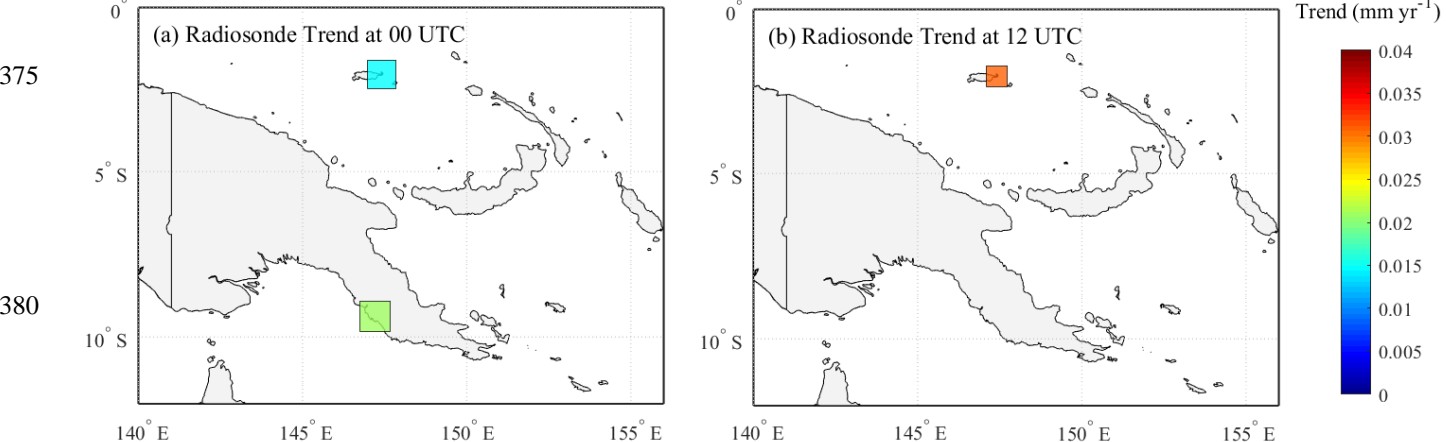

**Figure 5.** The estimated PWV trends ($mm\ yr^{-1}$) from the two radiosonde sites at their respective launch times. The size of the marker corresponds to the length of the PWV time series.

Figure 5 presents the estimated PWV linear trends of the two IGRA sites based on the monthly anomaly time series derived from radiosonde measurements. The marker sizes at each location are proportional to the time series (smaller square indicates shorter time series). The PWV trend estimates for radiosonde 92044 at Momote are 0.015mm $yr^{-1}$ (from 2002 – 2013) and 0.031mm $yr^{-1}$ (2007 – 2013) at 00 and 12 UTC respectively, as presented in Table 4. Similar upward trend is shown at radiosonde 92035 in Port Moresby at 0.021 mm $yr^{-1}$ for 00 UTC only from 2000 - 2013. The two radiosonde stations show upward PWV trends during their respective time series lengths, with only radiosonde 92044 indicating a statistically significant trend at both launch times. It can be emphasized that the potential influences of PWV trend estimates are due to sampling errors and huge interannual variations, usually linked to ENSO occurrences during this period (Wang et al., 2016).

The higher trend values estimated at radiosonde 92044 is due to a shorter time series. Estimated linear trends, including their uncertainties are generally higher for short time periods, and as the time period lengthens both parameter values will decrease. This is caused by the true short-term variation (the natural variability of the weather) which are not described by the model (Ning and Elgered, 2012). This area however, is beyond the scope of this study and will not be discussed.





**Table 4.** The estimated PWV trends (mm yr-1) at the two radiosonde sites with their corresponding Z-score. Trend values in boldface are denoted as statistically significant; Value with asterisk denotes that the trend is not statistically significant.

| | | 0000 UTC | |
|---|---|---|---|
| IGRA Station | Period | Test Score (Z) | Trend (mm yr$^{-1}$) |
| 92035 | 2000 - 2013 | 0.485 | 0.021* |
| 92044 | 2002 - 2013 | 2.453 | **0.015** |
| | | 1200 UTC | |
| | | Test Score (Z) | Trend (mm yr$^{-1}$) |
| 92044 | 2007-2013 | 1.981 | **0.031** |


The short/ different time windows observed in the radiosonde data within the study period time frame is caused by
significant deficiencies in the observing frequency and spatial density of radiosonde networks in PNG and the South Pacific region, with radiosonde observations in PNG being inactive since 2013. The local uncertainties in analysis are larger in areas with no conventional (non-satellite) upper air observations (Seidel et al., 2004). These sparse measurements of water vapour in the atmosphere has caused PWV trends and variability to not be well researched in the region. With GNSS becoming a valuable source of observation, a well-sampled and increased time series in radiosonde measurements over the PNG region is
necessary to verify the accuracy of regional spatial moisture trends and variations in the reanalysis, which will require further investigation in the future.

**4 GNSS PWV time series**

The three GNSS stations sample slightly similar climate conditions of the wetter lowland areas of PNG. Presented in Fig. 6, the 17 to 18 year absolute PWV estimates derived from these stations are plotted where available in the time series. The
moderate annual variation in PWV displayed by the locations reflect the relatively high humidity and evaporation rate over the coastal regions of PNG linking the influence of atmospheric-oceanic activities within the WPWP region.








**Figure 6.** The estimated PWV time series plots for the three PNG GNSS sites used in the study between 2000 to 2019.

455

The distinct peaks and dips in the time series occur approximately at the 'south-eastern' and 'north- western' monsoon seasons, however a lot of the variability is ENSO related (Trenberth et al., 2005). For instance, strong ENSO records such as the 2010/2011 La Niña event is evident in the timeseries and represented by significant drops in PWV by island stations PNGM and RVO around 2011. The range of PWV at the three sites are between 20mm and 80mm. The 17-18 year average and standard deviation of GNSS PWV is shown in Table 5. The average PWV values of the three coastal GNSS sites exhibit a relatively moist atmosphere and the standard deviation of the stations increases as their latitudes decrease. These values are typical for the South Pacific region.

465



**Table 5.** GNSS station positions with the mean GNSS PWV estimates and standard deviations (mm) over the 17 to 18-year period, respectively.

| GNSS station | Latitude | Longitude | Ellipsoidal height (m) | Mean (mm) | STD (mm) |
|---|---|---|---|---|---|
| LAE1 | -6.674° | 146.993° | 140.350 | 54.4 | 6.2 |
| PNGM | -2.043° | 147.366° | 116.343 | 54.8 | 5.9 |
| RVO_ | -4.191° | 152.164° | 266.642 | 52.9 | 5.9 |

## 5 Comparison with ERA-Interim PWV

As a widely used global reanalysis product for monitoring and studying the change of climate, the ERA-Interim reanalysis offers optimum humidity estimates. In this section, the ERA- Interim derived PWV time series and GNSS data are compared, including their seasonal cycles and PWV trends between the period of 2000 to 2019.

### 5.1 PWV comparison

The retrieval of PWV values at the GNSS sites from ERA- Interim reanalysis products follows the steps described in section 2.4, from 1 January 2000 to 31 August 2019 every 6 hours. The statistical comparison between the two datasets in presented in Table 6. The absolute differences (ERA-Interim minus GNSS) and the relative differences at the stations are calculated for the common observations (Zhang et al., 2017). The positive biases between ERA-Interim and GNSS are smaller than 1.5 mm showing an overestimation of ERA-Interim over coastal areas. This can be linked to the assimilation of Special Sensor Microwave Imager (SSM/I) data in the reanalysis, as the humidity is consistently higher in comparison to radiosonde measurements (Andersson et al., 2007).

**Table 6.** ERA- Interim PWV minus GNSS PWV comparisons.

| GNSS site ID | R | MnB (mm) | Mean relative PWV difference (%) | std (mm) | std relative PWV difference (%) | RMS (mm) |
|---|---|---|---|---|---|---|
| LAE1 | 0.78 | 1.46 | 0.78 | 4.39 | 8.46 | 4.62 |
| PNGM | 0.87 | 0.35 | 0.85 | 3.02 | 5.80 | 3.04 |
| RVO_ | 0.82 | 0.11 | 0.50 | 3.53 | 7.04 | 3.53 |
| Mean | | **0.64** | | | | **3.73** |





The relative differences are all smaller than 10%. The correlation coefficients (*R*) are above 0.75 at all stations, however these modest values can be related to the low number of high-quality measurements from conventional sources that are assimilated into the model from the region. The absolute standard deviations at the sites are between 3mm and 4.5mm and show that these are high precipitation regions, where high moisture and high moisture variability are observed. The average RMS error for all sites is 3.73mm respectively. This value can be comparable to the average RMS value of 3.82mm observed by Chen and Liu (2016) for GNSS - ERA-Interim comparison in tropical regions.

**Figure 7.** Comparison of the 6-hourly PWV estimates from ERA-Interim and GNSS. (a) regression plot with the correlation coefficient, (R) at station LAE1. (b) difference PWV time series (ERA-Interim – GNSS) (c) Comparison of the PWV anomalies (magenta: ERA-Interim, blue: GNSS). (d)-(f) and (g)-(i) as in (a)-(c), but for PNGM and RVO_, respectively.





The PWV monthly anomaly time series of GNSS and ERA-Interim data are derived following the method described in section 3.3. The PWV anomalies are positively correlated at all stations, exhibiting nearly similar ranges up to 10mm as presented in Figs. 7(c), (f), (i). The range of PWV anomalies depend on the mean annual water vapour content (Vey et al., 2009). The content of water vapour in the tropics is primarily influenced by changes in the global circulation patterns and to a

525 small extent affected by thermodynamical processes (Zveryaev and Allan, 2005). These global circulation patterns also act as the driving force for water vapour anomalies at an interannual timescale. The most influential global circulation patterns at interannual timescales is the ENSO, displayed in Figs. 7(c), (f), (i) by the significant troughs and peaks presented by both GNSS and ERA-Interim anomaly series. These features are reflective of the 1998-2001 and the 2010-2012 La Niña, and 2015-2016 El Niño events.

## 5.2 Seasonal Cycle

The seasonal cycle of PWV in the tropical regions is linked to atmospheric circulations causing the wet and dry seasons (Vey et al., 2009). Fig. 8 presents the seasonal cycle and interannual variability from GNSS and ERA-Interim at the three stations where the time series were the longest. Following Bock et al., (2007) the seasonal cycle is defined as the change in the

535 average of PWV for each month and the interannual variability as the standard deviation of monthly PWV. The agreement in PWV monthly averages between the two datasets is generally good.

The monthly PWV at the three stations show a rather weak seasonal cycle (6 to 10mm amplitude on average). Each station displays distinct features of PWV, with stations PNGM and RVO_ showing double peaks in mean PWV due to rapid water vapour changes. Station LAE1 at almost 7°S latitude presents one marked peak in mean PWV between December – March.

The double peaks for stations located close to the equator is observed in previous studies over their respective study regions e.g. Bock et al. (2007), Ssenyunzi et al. (2020), Lees et al. (2020).

Station PNGM and RVO_ display seasonal maxima in April/May and November/December, reflecting the two passes of the ITCZ annually and marks the transition between the south-eastern and north-western monsoon seasons. This is also marked by the interannual variability decrease of PWV at stations LAE1 and PNGM between June and July, reflecting the influence

of the prevailing south-east season. Station LAE1, however is evidently influenced by the two monsoon seasons in PNG, due to its location at the "mouth" of the north west- south east oriented Markham valley (McAlpine et al., 1983). The seasonal PWV maxima observed in April/May is higher than November/December at both island stations.





**Figure 8.** The monthly mean (top) and standard deviation (bottom) of PWV from GNSS (blue dashed lines with triangles) and ERA-Interim (yellow solid line with circles). GNSS PWV period for LAE1 is from 2001 to October 2003, 2008 to 2013 for PNGM, and 2014 to 2017 for RVO_. ERA- Interim PWV periods are matched with GNSS (adapted from Bock et al., 2007).



Although located on neighbouring islands, the inter-annual variability (bottom part of Fig. 8) at station RVO_ is very small while it is relatively pronounced at station PNGM during the austral summer. All stations display less variation of PWV in the wet seasons than in the dry seasons, which is linked to the relatively low rainfall events that occur during the dry months.

## 5.3 Diurnal Cycle

The close relationships between the diurnal variations of water vapour to many weather processes, such as moist convection, precipitation, convergence of near-surface winds, and surface evapotranspiration, provides a valuable feature of the physical parametrization in weather and climate models (Wang et al., 2002a; Dai et al., 2002). This fundamental mode of variability in water vapour is still not well studied, especially in the tropics due to the poor high temporal resolution datasets, which can be improved with GNSS observations (Yang and Slingo, 2001; Wang and Zhang, 2009). Presented in Fig. 9 are the

GNSS and ERA-Interim derived PWV diurnal cycles of the three sites. The representation of PWV anomalies show the monthly means (filled contours) and sub-monthly variability (standard deviation plotted as dashed contours).

The mainland station LAE1 is marked by a pronounced 24 hr diurnal oscillation, showing a morning maximum between 0800 UTC and 1000 UTC and a minimum in the evening between 0000 UTC and 0200 UTC. The diurnal variation of the peak is more pronounced during the north-western monsoon season (December – March), reaching -2.6mm to 3.6mm, compared to

the south east season. This may be due to the enhancement of the well-developed land circulation system during this period. The large amplitudes in PWV over this region is a result of conditionally unstable air with high moisture content, reflecting the deep convection activities in the Huon Gulf area in conjunction to the timely presence of a branch of ITCZ through the region (McAlpine et al., 1983). In general, the more intense the deep convection, the stronger its diurnal variation (Gray and Jacobson, 1977; Tian et al., 2004).

The diurnal variation at LAE1 is also agreeable to the daily cycle of convection/precipitation in the tropics, with maximum convection/precipitation occurring in the late afternoon and evening, between 1700 UTC and midnight, in association with the land-sea breeze effect (Yang and Slingo, 2001). This also demonstrates the high incidence of rain at night during January compared to July, which is considerably less marked. In comparison, ERA-Interim underestimates the diurnal amplitude and does not accurately capture the variations of PWV during the austral winter. However though, similar to GNSS, the variability

is at maximum during the evening, which might reflect strong convective activity.

On Manus Island, GNSS derived PWV at station PNGM also displays marked diurnal variations with seasonal dependency. The diurnal amplitudes range between a maximum of ±1.2mm. During the transition phase (between March and April) from north-westly (austral summer) to south-easterly (austral winter) season, the diurnal cycle observed corresponds to that at LAE1 suggesting that local circulations are more dominant during this period. The maximum anomalies are observed between 0400

UTC and 1200 UTC from January to June and during the evening between 2000 UTC and 2200 UTC from July to October. The strong variability in PWV is evident during the south easterly season.






**Figure 9.** Monthly mean of PWV diurnal anomaly at GNSS stations are of similar periods to Fig. 8. (a)-(c) for GNSS data and (e)-(f) for ERA-Interim at stations LAE1, PNGM and RVO_. Shadings between red and blue presents the monthly diurnal averages (refer to color bars below each plots) and dotted contour lines indicate the sub-monthly standard deviations (2mm interval).





The observed diurnal features in seasonal change of PWV maximum could likely be linked to regional circulation pattern changes over the New Guinea islands of PNG, where physiographic barriers of the islands only minimally impede the broadscale flow. The strong north west trade winds in January intensify during the day, and become less well-developed during the transition period in April. Between June-July, the more freshly moist south-eastly winds of similar strength throughout the course of the day supersede the north-west components. By October, the weakening flow of the south-easterlies is interchanged by the cyclic strengthening of the north-westerly component, returning to the January situation (McAlpine et al., 1983). For PNGM, ERA-Interim is unable to reproduce the observed seasonal shift and variability in PWV, and presents a morning peak between 0900 UTC and 1200 UTC throughout the year.

Station RVO_ on the island of New Britain demonstrates diurnal PWV variations almost similar to station PNGM. The small amplitudes of PWV correspond to the smaller variations in the precipitation frequency during the day, or between seasons or intensities over Rabaul. The minima are observed in the afternoon, and shows a slight peak in PWV around 1500 UTC compared to station PNGM during the austral winter. This feature may also be associated with the movement of ITCZ over the region. The maximum peaks between 0600 UTC and 1000 UTC during December and March (austral summer) demonstrates the significance of the north-westerly flow over Rabaul, which is enhanced due to the slight physiographic barrier of the area. The strong variability observed during evening and night time in November-December indicates the strengthening of the moister north westerlies over Rabaul. At this location, ERA-Interim diurnal amplitude and phase are underestimated, especially during the austral winter.

### 5.4 GNSS and ERA-Interim PWV trend comparison

The GNSS and ERA-Interim PWV linear trends are derived from the monthly PWV anomalies presented in Figs. 7(c), (f), (i), following the method described in section 3.3**Error! Reference source not found.Error! Reference source not found.**. GNSS trends are estimated for the time series of 17 to 18 years long, whilst ERA-Interim are for the period of 2000 -2019.

Stations PNGM and LAE1 show significant PWV tends larger than 0.01mm yr$^{-1}$ at 0000 UTC respectively. The upward trend at station PNGM agrees with radiosonde 92044 over Manus island in section 3.3. Only station RVO_ shows an insignificant decreasing trend on the eastern region of PNG. For ERA-Interim, all stations reproduce a statistically significant upward trend in PWV. These results are similar to radiosonde data mainly due to unhomogenized radiosonde humidity records that is assimilated in the reanalysis. The comparison between GNSS and ERA-Interim in Table 7 show the consistency in statistical significance of PWV linear trends estimates at 0000 UTC and at 1200 UTC.



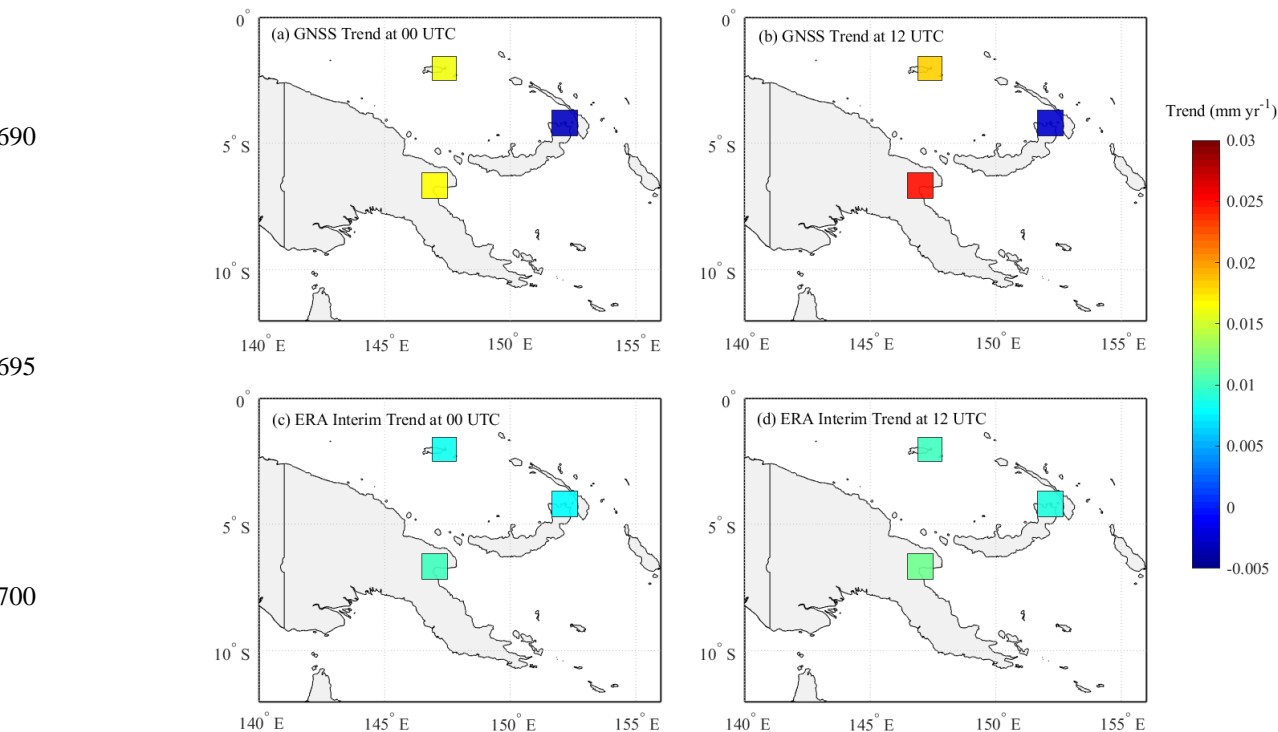

**Figure 10.** The geographical distribution of estimated PWV trends (mm yr$^{-1}$) at (a)-(c) 0000 and (b)-(d) 1200 UTC. GNSS period is at 17-18 years and ERA-Interim period is at 19 years (2000 -2019).

The positive trends of PWV from GNSS agree in sign to the global upward trends reported by Trenberth et al. (2005), reflecting the expected global change in atmospheric moisture due rise in temperatures (Trenberth et al., 2003; Zhang et al., 2013). Although governed by the Clausius-Clapeyron equation of about 7% K$^{-1}$, the PWV trends are smaller suggesting constant relative humidity (e.g. Soden et al., 2002). These regional trends of PWV are not simply linear and may vary over different periods, but provide support for future research in understanding the variations in moisture and precipitation trends over PNG region.

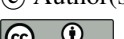



**Table 7.** The estimated PWV trends (mm yr⁻¹) based on GNSS (17 to 18 year period) and ERA-Interim PWV (19 year period) at all stations. The values in boldface denote statistical significance with 95% confidence level; values with asterisks are not statistically significant. Z-score is also presented.

| | | GNSS | | | |
|---|---|---|---|---|---|
| Station | Test Score (Z) | 0000 UTC (mm yr⁻¹) | Test Score (Z) | 1200 UTC (mm yr⁻¹) | All Times (mm yr⁻¹) |
| LAE1 | 3.279 | **0.017** | 2.947 | **0.025** | **0.026** |
| PNGM | 5.167 | **0.016** | 5.162 | **0.018** | **0.017** |
| RVO_ | -0.797 | -0.003* | -0.527 | -0.003* | -0.004* |
| | | ERA-Interim | | | |
| Station | Test Score (Z) | 0000 UTC (mm yr⁻¹) | Test Score (Z) | 1200 UTC (mm yr⁻¹) | All Times (mm yr⁻¹) |
| LAE1 | 4.244 | **0.009** | 3.731 | **0.012** | **0.011** |
| PNGM | 4.473 | **0.008** | 5.070 | **0.009** | **0.009** |
| RVO_ | 3.240 | **0.008** | 3.708 | **0.009** | **0.009** |

**6 Analysing GNSS PWV variability during ENSO events**

A correlation analysis is applied in this section to analyse the relationship between GNSS PWV variability and weekly SSTa during two major ENSO events in PNG between 2010 and 2016, with all coefficients significant at a 99% confidence level (Suparta et al., 2017). The correlation between the two variables is studied when the SSTa at the Niño 3.4 and Niño 4 regions is >±0.5°C. The increase intensity phase refers to the SST value increasing/decreasing during the El Niño/ La Niña
730 event until it reaches the SST peak of the event, while the decrease intensity phase explains the opposite by passing the ENSO threshold (±0.5°C) back to the neutral phase.

**6.1 Bimodal La Niña Event (2010 – 2012)**

The "double-dip" La Niña event between 2010 to 2012 consists of four phases, i.e. the two decrease phases (increase intensity) and two increase phases (decrease intensity) of the SSTa variability in the Niño 3.4 and Niño 4 regions.

**6.1.1 First case (June 2010-May 2011)**

Figure 11 presentsError! Reference source not found. the relationship between GNSS PWV and SSTa variability during the first case of the 2010/2012 La Niña event. As the event intensified in SSTa values, LAE1 displayed no correlation while stations PNGM and RVO_ displayed negative moderate relationships at -0.373 and -0.279 for the Niño 3.4 and -0.414 and -0.244 for the Niño 4 regions respectively. As shown in Table 8, the better relationships observed with Niño 4 region compared

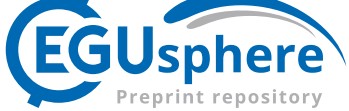

to the Niño 3.4 region is influenced mostly by its more western position to PNG. Although the relationship trend observed between GNSS PWV and SSTa is positive, the correlation coefficients obtained are negative indicating the cold phase. This is also conclusive to the low amounts of PWV in the atmosphere during La Niña over the region as most of the PWV are expected to precipitate as rain (Suparta and Iskandar, 2012).

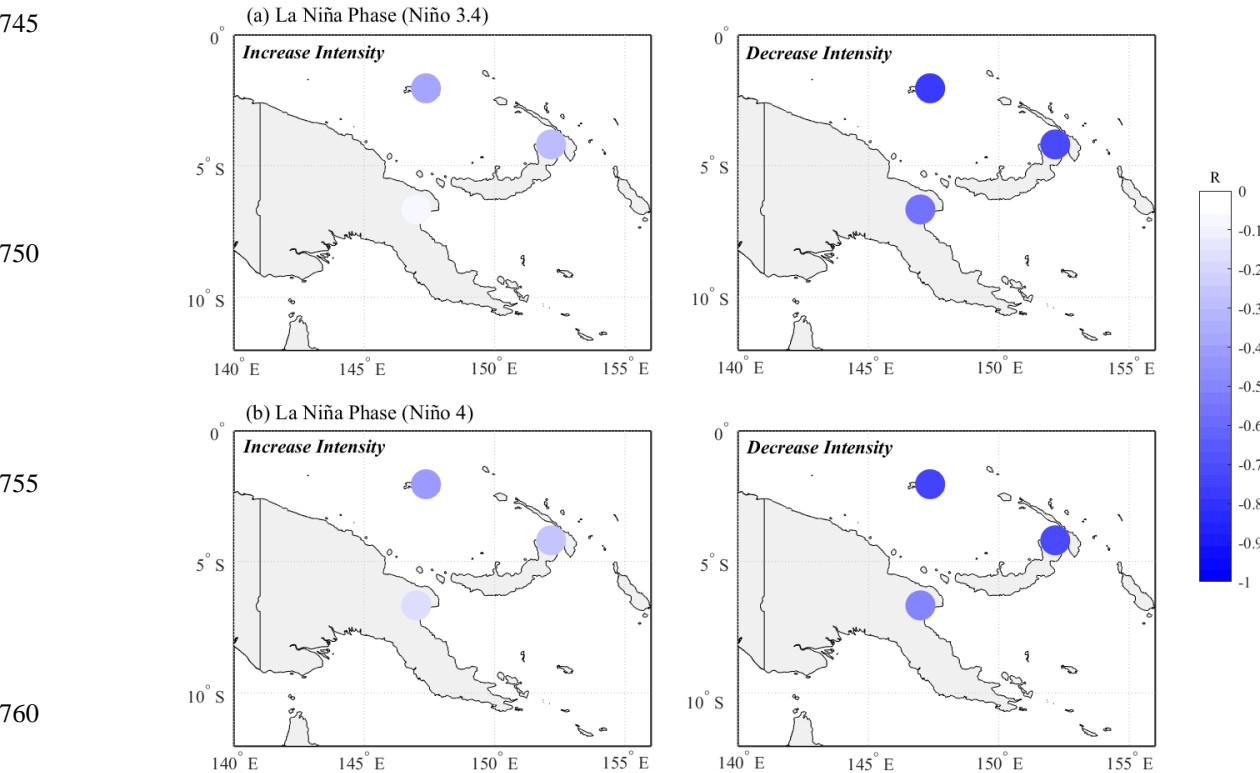

**Figure 11.** The geographic representation of the relationship between GNSS PWV and SSTa during the two intensity phases of the 2010/2011 La Nina event at (a) the Niño 3.4 region and (b) the Niño 4 region.

As the intensity of the event decreased from October 2010 to May 2011, both island stations displayed strong relationships, with PNGM having the strongest relationship at -0.78 and -0.74 for both regions. This may have highlighted the drought period experienced over the island region of PNG during this event instead of the general expectation for above average precipitation (Smith et al. 2013). Station LAE1 displayed more moderate relationships at -0.57 and -0.49 respectively.





**Table 8.** The correlation coefficients between GNSS PWV and SSTa during 2010/2011 La Niña event. (-) denotes no correlation.

| | Increase Intensity | | Decrease Intensity | |
|---|---|---|---|---|
| Station | Niño 3.4 | Niño 4 | Niño 3.4 | Niño 4 |
| LAE1 | - | -0.154 | -0.574 | -0.487 |
| PNGM | -0.373 | - 0.414 | -0.778 | -0.744 |
| RVO_ | -0.279 | -0.244 | -0.709 | -0.718 |

**6.1.2 Second case (August 2011-February 2012)**

The re-occurrence of La Niña between the austral spring and summer period of 2011/2012 was again characterized by colder-than-average sea surface temperatures in the equatorial Pacific Ocean. Although the event had weak-to-moderate La Niña conditions, the warmth of global temperatures during the second part of 2012 was heavily influenced by the early- year La Niña, making it the third warmest La Niña year on record (Osborne and Blunden, 2013).

**Table 9.** The correlation coefficients between GNSS PWV and SSTa during 2011/2012 La Niña event.

| | Increase Intensity | | Decrease Intensity | |
|---|---|---|---|---|
| Station | Niño 3.4 | Niño 4 | Niño 3.4 | Niño 4 |
| LAE1 | -0.433 | 0.699 | -0.431 | -0.062 |
| PNGM | -0.546 | 0.464 | -0.36 | -0.261 |
| RVO_ | -0.538 | 0.538 | -0.256 | -0.356 |

From Fig. 12, the increase intensity phase showed that all stations displayed weak to moderate negative relationships for the Niño 3.4 region. However, all three stations had moderate to relatively strong positive relationships for the Niño 4 region as the event intensified, with the mainland station LAE1 displaying the strongest at 0.69 respectively (Table 9). These positive relationships reflected the suppression of convection near the dateline (within the Niño 4 region) in association with La Niña



(CPC, 2011). This might have prolonged the event to peak in SST values on January 2012 in this region. As the intensity of the decreased, all stations displayed weak relationships at both Niño regions, indicating the event's passing.

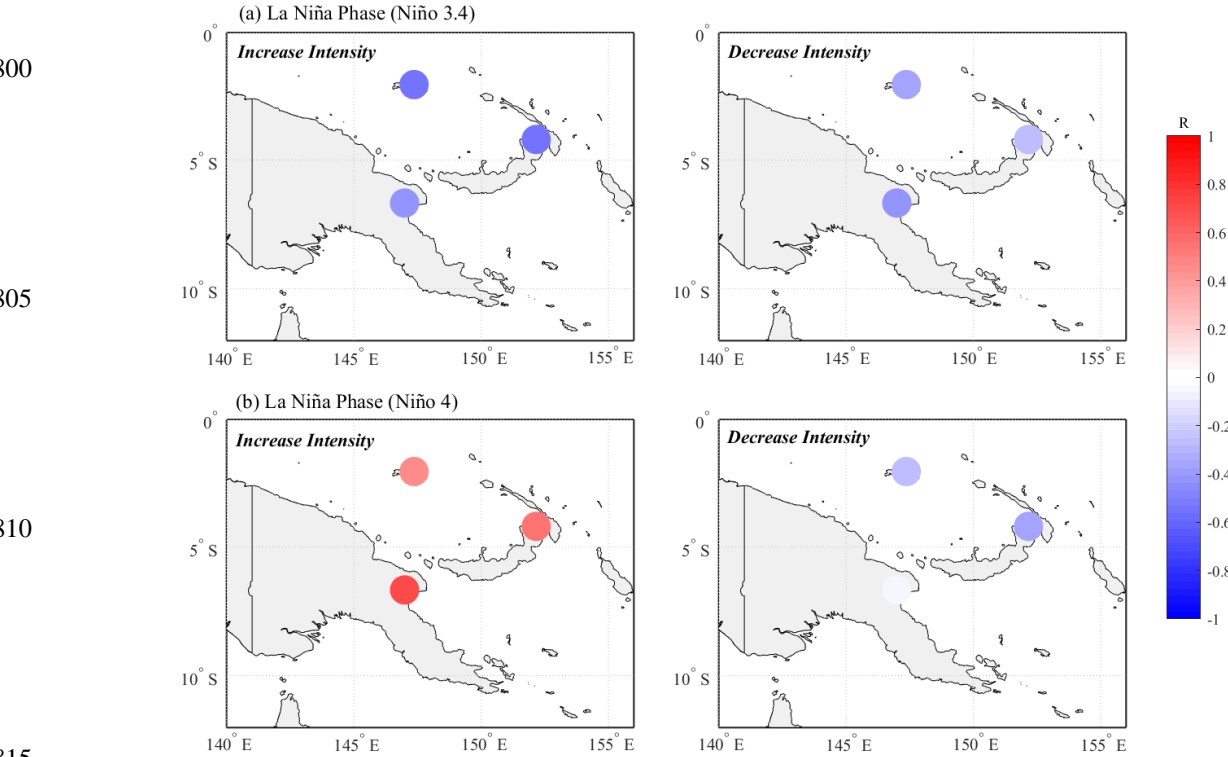

**Figure 12.** The geographical representation of relationship between GNSS PWV and SSTa during the two intensity phases of the 2011/2012 La Nina event at (a) the Niño 3.4 region and (b) the Niño 4 region.

**6.2 El Niño Event (April 2015 to April 2016)**

On November 2015, the 2015/2016 El Niño event displayed a record peak of 3°C in the Niño 3.4 region, surpassing the previous record of 2.8°C set in January 1983. The Niño 4 region achieved a peak at 1.6°C, as the average conditions were already warm. In Fig. 13, the relationship between GNSS PWV and SSTa for the Niño 3.4 region during the intensity increase phase at all three stations was very weak, with LAE1 showing no relationship.




**Table 10. The correlation coefficients between GNSS PWV and SSTa during 2015/2016 El Niño event. (-) denotes no correlation.**

|  | Increase Intensity | | Decrease Intensity | |
| --- | --- | --- | --- | --- |
| Station | Niño 3.4 | Niño 4 | Niño 3.4 | Niño 4 |
| LAE1 | - | 0.457 | -0.409 | -0.467 |
| PNGM | -0.272 | -0.104 | - | - |
| RVO_ | -0.387 | - | -0.384 | -0.414 |


However, for the Niño 4 region LAE1 displays a moderately positive effect in PWV compared to the two island stations which displayed weak relations. This could indicate the more dynamic transfer in atmospheric water vapour around the mainland region through low level westerlies and tropical convection shift in response to El Niño during the north west

monsoon season. As the intensity of El Niño decreased, PNGM displayed no correlation while LAE1 and RVO_ both displayed moderate to weak negative correlation at both regions, as shown in Table 1.

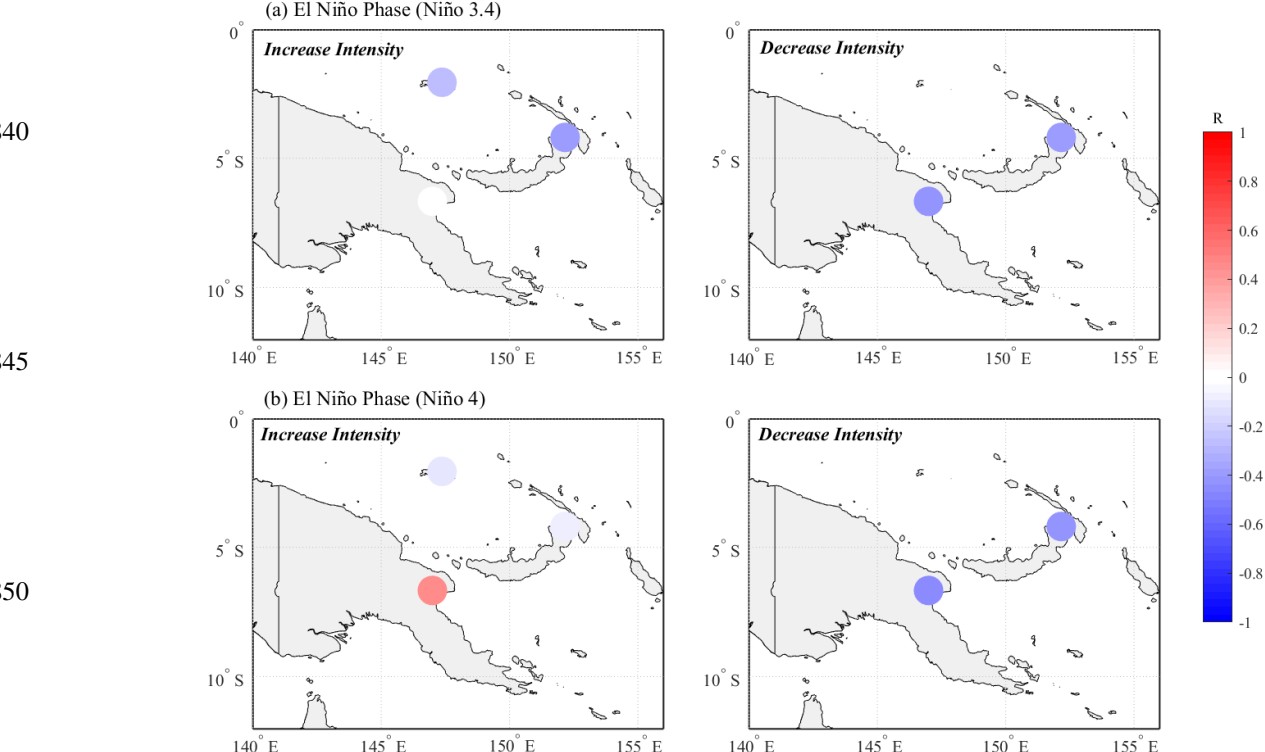




**Figure 13.** The geographical representation of relationship between GNSS PWV and SSTa during the two intensity phases of the 2015/2016
El Niño period at (a) the Niño 3.4 region and (b) the Niño 4 region.





## 7 Discussion

Using archived historical GNSS datasets and increasing the number of operational GNSS stations provides a resourceful opportunity to study water vapour climatology, especially over the PNG region with currently inactive radiosonde stations. Particularly in the tropics, biases in GNSS-derived PWV with respect to radiosonde RS80-15G model (Ciesielski et al., 2003),

RS80A, RS80H, and RS90 models (Wang et al., 2007), and RS92 model exhibited a dry (negative) bias in radiosonde measurements (Wang et al., 2013). This was significantly evident in the earlier "Tropical Ocean and Global Atmospheric/Coupled Ocean-Atmosphere Response Experiment (TOGA COARE)" radiosonde observation campaign around the tropical western Pacific warm pool region (Wang et al., 2002b). However, Fernández et al. (2010) showed both moist (positive) bias and dry bias when comparing between Vaisala RS80 radiosonde measurements and GNSS over Argentina. This

varied between the different launch sites and the different models used for estimating $T_m$. Comparably, Mengistu et al. (2015) underlined the essential use for surface pressure observations to derive GNSS measurements of PWV by comparisons with RS92, and demonstrated improvements in bias and RMS compared to the use of the GPT model which had presented significant moist bias. The uncertainty in ERA-Interim weighted mean temperature and pressure used to derive GNSS PWV may have some role to the overall discrepancies between radiosonde and GNSS PWV, deducing wet (positive) bias in Fig. 3.

Therefore, more robust radiosonde and GNSS campaigns using observed surface pressure and temperature is needed to quantify the nature of the bias over the region.

The accuracy of the RS80-15GH model used at Momote may also be questionable due to the design age as no cross comparison has been done before. Therefore, careful test on of the Vaisala radiosonde is necessary to solve the reason for the bias and fix the problem with appropriate correction factors, as reported by Wang et al. (2002b). This task may require further

investigation, however is beyond the scope of this research. The PWV trends presented in this study at the three station, although shorter, are significant because there are no artificial discontinuities in the data that can query their accuracy, compared to radiosondes. It further supports climate trends investigations into the unknown diurnal asymmetry of PWV trends due to its long-term stability and high temporal resolution (Wang et al., 2016).

GNSS data can also be used to assess moisture fields performances in the ERA-Interim reanalysis product by serving as

an independent source, due to its insensitivity to rain and clouds. The relationship differences between GNSS and ERA-Interim PWV at the three stations varies between different seasons and diurnal variation amplitudes, which depends on the geographic location and the topographic features of the region. This is demonstrated by the wet bias observed at station LAE1, located further inland compared to the stations RVO_ and PNGM at more lowland coastal areas. The accuracy of PWV anomalies from GNSS also show that ERA-Interim represents the interannual variations in PWV correctly over the three sites. In most

cases in the GNSS PWV time series, the water vapour is underestimated by the model. Furthermore, PWV anomaly for stations PNGM and RVO_ are underestimated by the model during the record breaking La Niña event of 2010 to 2012. In addition, stations LAE1 and PNGM are overestimated by the model during 2015/2016 period, which is characterized by the strong El Niño event.





Although predicting ENSO events allows the relationships between tropical Pacific SST and global and/or regional
precipitation to be detected, there is inconsistency between ENSO and regional rainfall relations (Zhu, 2018). It is also still not
clear what the mechanisms conveying the remote influences of ENSO over PNG are and how it affects local precipitation
especially over the New Guinea islands region of the country. This case is demonstrated by the strong negative relationship
between GNSS PWV and SSTa at stations PNGM and RVO_, which reflected drought by the decrease in water vapour
corresponding to the rainfall deficiencies experienced over the island region during this period. The same evaluation is
observed for El Niño events, as no individual ground-based station will necessarily reflect the impacts of ENSO over such a
large geographically complex island country.

**8 Conclusion**

The transmission of radio signals from GNSS satellites to the ground are delayed by the atmosphere. The delay caused by
the water vapour component of the troposphere provides the opportunity for the retrieval of PWV using ground-based GNSS.
In this study, the PWV over the PNG region is determined using GNSS with comparisons to radiosonde and the ERA-Interim
reanalysis. The GNSS stations are part of different regional networks in PNG and their data are in the period between 2000 to
2019. ZTD measurements for PWV calculations are processed and solved in the PPP mode using reprocessed ES2 and ESA
satellite and orbit clock products in the IGS08 reference frame for the whole period. This is to avoid contaminating the long-
term signals and any inconsistencies among the different reference frames.

The weighted mean temperature $T_m$ and surface air pressure $P_s$ values derived from ERA-Interim are used to derive GNSS
PWV, due to lack of colocated meteorological data. $T_m$ values derived from ERA-Interim are compared with radiosonde 92044
over Manus island, with an assessed accuracy of approximately 1.1 K. Validating the quality and consistency of GNSS PWV
derived at station PNGM with radiosonde 92044 dataset found a mean bias of 1.7mm and RMS of 3.5mm. The impacts of the
characterized error sources on data taken over Manus island exhibits confidence in the reliability and robustness of the GNSS
derived PWV, for further analysis with ERA-Interim reanalysis to fulfil the purpose of this study. This also fills the void in
investigating water vapour variability using ground-based GNSS over PNG region for the first time, particularly along the
coastal regions. In addition, future works is still required to establish fitting relation of error sources over the PNG region by
use of surface air temperature and pressure observations for GNSS PWV retrieval, together with statistical analysis with routine
radiosonde campaigns.

Knowledge of the long-term trends in PWV provides crucial information for weather studies, and was estimated using the
Mann-Kendall tau method and Sen's nonparametric method. An attempt to estimate PWV trends from radiosonde observations
was performed accordingly with respect to the launch times. Estimated PWV trends at radiosondes 92035 and 92044 are
positive for time series that are 6 to 13 year long, with radiosonde 92044 estimated PWV trends being statistically significant
at both launch times. Because the launch time windows at the two radiosonde sites are not concurrent (for launch times as
well), no specific conclusion is drawn about the mean trend over the research region from radiosondes measurements.



More reliably as an independent data source, GNSS is used for comparison with ERA-Interim to study the spatial-temporal variation of PWV around the coastal regions of PNG. The biases in ERA-Interim PWV relative to GNSS is smaller than 1mm at the two island stations and the average RMS for all stations is 3.73mm. The seasonal variability of PWV between GNSS and ERA-Interim are generally unique but in good agreement at a larger timescale. The two monsoonal characteristics over the PNG, which is predominately influenced by the continuous passing of the ITCZ is well reflected in the seasonal variability of PWV. The mainland station LAE1 presents the most significant seasonal cycle with monthly-mean ranging from minima values of 48mm in September to maxima of ~58mm in February.

In addition, larger diurnal PWV amplitudes are observed over the mainland (~ 3-4mm for LAE1) compared to the island stations (~1mm for PNGM). All stations exhibit daily maximums occurring in the morning and daily minimum occurring in the late afternoon, however station PNGM presents a distinct seasonal modulation phase in PWV diurnal cycle with maximum PWV occurring in the morning during the north-west season (around the austral summer), but in the late evening during the south-east season (austral winter). All stations experience sub-monthly variability peaks in the evening which might be associated with the maximum convection activity over land. The diurnal cycle in ERA- Interim PWV shows significant differences to GNSS at the three sites, notably by the underestimation of the PWV amplitudes at a diurnal scale. It is also not able to capture the transitioning shifts between the monsoon seasons, further confirming the global model deficiencies in diurnal cycle diagnostics (Wang and Zhang, 2009). Although these results provide confidence on the quality and potential of GNSS PWV estimates over particularly the coastal regions of PNG, further use of ground-based GNSS is essential to observe and predict the spatial and temporal evolution of water vapour at various local time scale settings.

Extending PWV trend analysis for GNSS time series lengths at 17 to 18 years show absolute values of PWV trends at less than 0.03mm yr$^{-1}$ for all stations, with insignificant negative PWV trend at station RVO_. Similarly, ERA-Interim results show statistically significant increasing PWV trends for all stations from 2000 to 2019.

## 8.1 ENSO Events

The response of PWV to ENSO events is investigated by analysing the relations between GNSS PWV and SSTa variability during two significant ENSO events (2010/2012 La Niña and 2015/2016 El Niño) that impacted PNG in the last decade using a correlation analysis. A strong negative relationship is displayed at the island stations (PNGM = -0.778 and RVO_ = -0.709) compared to the mainland station LAE1 (R=-0.574) for the Niño 3.4 region. These high negative coefficients reflect the severe drought experienced over the island region, due to rainfall deficiencies contrary to the general expectation for above average rainfalls during the event, which is readily explained by the regional rainfall shifting westward in response to La Niña conditions (Smith et al., 2013).

By comparison, the 2015/2016 El Niño event displayed weak negative correlation at PNGM while RVO_ and mainland LAE1 observed moderate negative correlation for both Niño regions. This also indicates the decrease in PWV content over the island regions and the mainland, which is associated with dry conditions during El Niño. This event severely affected 2.5 million people in the Highlands regions of PNG, with similar effects also observed in other South Pacific island countries as





well as Australia and New Zealand. The variation of PWV during these two ENSO events also demonstrated the relationship

between PNG rainfall and ENSO indices, indication that only strong ENSO events will have a clear impact.

*Data availability.* IGNSS datasets are available at the NASA CDDIS (https://cddis.nasa.gov/Techniques/GNSS/ (Noll et. al, 2009). The weekly OISST v.2.1 SSTa data can be obtained from NOAA ( https://www.cpc.ncep.noaa.gov/data/indices, last access: 05 April 2021, (Reynolds et al., 2002). The IGRA radiosonde data archived at the National Climatic Data Centre of NOAA can be accessed through the GMET Online Service tool (http://gmet.users.sgg.whu.edu.cn/en/ (Zhang et al., 2017)

which is in care of the GNSS Research Centre of Wuhan University, and ERA-Interim products are archived at https://apps.ecmwf.int/datasets/data/interim-full-daily/ (Dee et al., 2011).

*Author Contributions.* YL, CS and WZ designed the study. WZ oversaw the research and provided invaluable suggestions and comments throughout. AS ran the processing and performed the data analysis. JB maintained operations of the GMET website. AS prepared the paper, with contributions from all authors.

*Competing interests.* The authors declare that they have no conflict of interest.

*Acknowledgements.* The authors would like to thank Steve Saunders of the Rabaul Volcano Observatory for providing GNSS datasets of the Rabaul Caldera network. We are also grateful to Kasis Inape and Kisolel Posanau of the PNG National Weather Service for providing monthly meteorological data for comparison purposes. Author (A. Senat) was supported by the China Scholarship Council (2019276190003), which is gratefully acknowledged.

*Financial Support.* This work has been supported by the National Natural Science Foundation of China (41961144015; 42174027), the Key Research and Development Program of Guangxi Zhuang Autonomous Region, China (2020AB44004) and the Fundamental Research Funds for the Central Universities (2042022kf1198).

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
