# Peer review of "The use of ground-based GNSS for atmospheric water vapour variation study in Papua New Guinea and its response to ENSO events"

_EGUsphere, 2022_

## Referee Comment (RC1)

**Review of "The use of ground-based GNSS for atmospheric water vapour variation study in Papua New Guinea and its response to ENSO events."**

**General comments**

The manuscript promotes the use of ground-based GNSS time series (of about 20 years) at 3 sites in Papua New Guinea to study in particular the temporal variability of PWV at this important region (West Pacific Warm Pool) in terms of global water vapour distribution. In conjunction with GNSS PWV retrievals, ERA-interim PWV output and radiosonde observation are used.

Overall, I certainly strongly encourage the use of GNSS PWV retrievals for atmospheric applications, to bridge the gap between the geodetic and atmospheric science communities. However, I would not rate the present manuscript as a good example in this sense.

To start with, the "literature overview" given by the authors in the introduction to describe the state-of-the-art of ground-based GNSS PWV retrievals used for atmospheric applications is really outdated (just have a look at the publication years of the cited references!). The authors should consult the following review paper (and many more recent references therein) to be updated about recent results and applications: Vaquero-Martinez, J.; Anton, M. Review on the Role of GNSS Meteorology in Monitoring Water Vapor for Atmospheric Physics. Remote Sens. 2021, 13, 2287, https://www.mdpi.com/2072-4292/13/12/2287. See also a "Climate Modelling and Monitoring Using GNSS" Special Issue in the same journal: https://www.mdpi.com/journal/remotesensing/special_issues/Global_Climate_GNSS. Since you submitted to ACP, you should also have been aware of the following ACP special issue https://acp.copernicus.org/articles/special_issue400_89.html on "Advanced Global Navigation Satellite Systems tropospheric products for monitoring severe weather events and climate", in which you also find some older references that really can help you in writing a more up-to-date introduction on the use of GNSS PWV for climate applications. By the way, the observation of PWV from space also has a lot more evolved than written in your introduction (e.g. between lines 32-44), see e.g. the recent ACP Special Issue (https://acp.copernicus.org/articles/special_issue1118.html). In this SI, you also find a reference (Wagner et al., 2021) to the study updating the Wagner et al. (2005) paper you are referring to (two times in lines 80-99) for illustrating past research on the response of GNSS PWV to ENSO events. Please also update this description. Also, the assimilation of water vapour observations in numerical weather prediction models has a lot more evolved than has been written down in the manuscript (see e.g. the evaluation of PWV of 5 reanalysis products with GNSS in Wang, S.; Xu, T.; Nie, W.; Jiang, C.; Yang, Y.; Fang, Z.; Li, M.; Zhang, Z. Evaluation of Precipitable Water Vapor from Five Reanalysis Products with Ground-Based GNSS Observations. Remote Sens. 2020, 12, 1817. https://doi.org/10.3390/rs12111817). And also the description of the climatology of the study area (section 2.1) is really based on very old references (before 2000!). Is this information still valid given the recent climate change?

Another example of using "outdated" resources is the use of ERA-interim. As you should know, the current state-of-the-art reanalysis product of ECMWF is ERA5 (fifth generation as opposed to the third generation ERA-interim). The authors should at least provide some explanation why they still use ERA-interim. ERA5 has clear advantages over ERA-interim: (i) a much finer horizontal resolution (30 km grid

vs. 80 km grid of ERA-interim), which is in particular important for comparison with coastal/island sites due to spatial representativeness arguments, and (ii) a temporal resolution of 1 h (instead of 6 h for ERA-interim), extremely important when assessing the diurnal PWV variability (one of the aims of this study). Of course, also the physics/parameterizations in the numerical weather output model and the data assimilation have been majorly improved with respect to ERA-interim. Therefore, I don't see any reason for using ERA-interim instead of ERA5 and I would really urge the authors to include ERA5 in this study.

In particular, the course time resolution of ERA-interim (and GNSS PWV, as ERA-interim is used as the source for $T_m$ and $P_s$) really undermines the analysis of the diurnal cycle in Sect. 5.3. In addition, the contour plots in Fig. 9 (with times marked on the axis every 2hs) are really misleading, as the contours are to a large extent filling up the observations at 0h, 6h, 12, and 18 UTC. Therefore, I would describe this entire section, based on this figure, rather "tentative". Only in the figure 9, the connection between the local time and UTC time is made, but this should also be included in the discussion, because it's not quite obvious if "afternoon" and "evening" (e.g. in line 615: "evening between 2000 UTC and 2200 UTC") are really falling in that part of the day in terms of local time.

The analysis using the radiosonde measurements is also very chaotic and misleading! It is mentioned that radiosonde station at Momote airport has been type Vaisala RS80-H (do not be mistaken with RS80-A, which has another humicap sensor!) over the entire time period, which is known to suffer from a dry bias (lines 197-198), but the apparent moist bias of the radiosonde w.r.t. GNSS PWV is explained "as moist biases are known to exist for large PWV ranges in Vaisala radiosonde types" (line 343). And, as only one type of radiosonde has been used at Momote through the observation time period, the argument that "radiosondes are evaluated to having its own systematic bias depending on the types of radiosonde and should not be assumed to provide certainty when assessing PWV intertechnique biases" (lines 309-310) does not really make sense. Instead, I would expect an explanation why there doesn't seem to be any biases between radiosondes and GNSS in the early years (before 2005, see Fig. 3), but a moist bias after 2006 onwards. And why do the authors make the distinction between 0h and 12h UTC measurements? I guess it has to do with the known radiation dry bias for some of the radiosonde types (right?), but then you should explain and mention which are the daytime and nighttime measurements.

The section about the $T_m$ error analysis does not bring any added value if you do not quantify the impact $T_m$ has on the PWV. Using the formulas used for the GNSS ZTD to IWV conversion, the impact of the surface pressure on the IWV is largest: a 1 hPa change in Ps gives an IWV change of 0.36 mm, whereas a 1 K change in Tm leads to an IWV change in the range 0.05 to 0.20 mm, depending on the ZTD and Ps values. So, you should also describe the impact of Ps and the altitude offset correction between the model grid height and station height. And more importantly, you are using the weighted mean temperature and surface pressure from ERA-interim to convert the GNSS ZTD to PWV, and then you are comparing the GNSS PWV with the ERA-interim PWV. To which extent can these two PWV datasets then considered as independent from each other? This question needs to be commented!

Another point that I want to make is about the calculation of PWV "long-term" trends based on time series of 7, 13 (radiosondes) and almost 20 years. How reliable and meaningful are these? In the cited (!) publications of Alshawaf et al., it is concluded that the number of years required to detect significant PWV trends varies between 30 and 40 years, if we take the autocorrelation of the PWV time series into account (according to Weatherhead, E.C.; Reinsel, G.C.; Tiao, G.C.; Meng, X.-L.; Choi, D.; Cheang, W.-K.;

Keller, T.; DeLuisi, J.; Wuebbles, D.J.; Kerr, J.B.; et al. Factors affecting the detection of trends: Statistical considerations and applications to environmental data. J. Geophys. Res. 1998, 103, 17149–17161). So, some caution is really needed when estimating "trends" from those very short time series, and comparing those trends with each other (e.g. between RS and GNSS with different periods, between GNSS and ERA-interim, between 0h and 12h UTC). What is the physical meaning/interpretation of those trends? In particular, the section analyzing the ENSO events clearly demonstrates that the interannual variability is the main driver of the PWV variability on these short time scales (NOT climatological time scales!).

This brings me to my last point of major criticism: the regression analysis between PWV and ENSO proxies during ENSO events. I'm not very convinced by its presentation and even the obtained results. Of course, from the PWV time series plots in Figs. 6 and 7, it is clear that the PWV behavior around 2010-2011 and 2015-2016 is driven by ENSO and a closer look is needed to study this connection. But looking at Fig.2, is it really needed to use two different SSTa regions and two ENSO period definitions for making this linkage, if there are only minor differences between those two and the geographical component of the two regions is not substantially taken into account in the interpretation of the correlations? Also the distinction between "increase intensity" and "decrease intensity" phases of the ENSO events does not seem really scientifically funded, besides increasing the significance of the correlations during specific phases (because insignificant during the entire ENSO event?). Please give a better rationale why making this distinction between those different phases. I also find the Figs. 11-13 not very instructive. Instead, I would be more interested in a plot showing the regression plots themselves (or at least the time series of weekly SSTa and weekly (right?) PWV values → is the weekly time scale the most appropriate?). Also, in the discussion, quite often is referred to precipitation/rainfall characteristics of the region to explain the PWV and ENSO linkage, but not a single precipitation/rainfall dataset in this region is actually included in the (regression) analysis. Are such datasets not available? Please comment.

Finally, the discussion and conclusion sections are badly written, in the sense that they don't put the obtained results in a larger perspective, as would be expected in a discussion section, and the conclusion contains too much information that merely belongs to an introduction section (e.g. first paragraph), and does not distillate the most important findings from trivial ones. However, as both the analysis and the manuscript need a major revision, I won't go into details here.

---

## Author Comment (AC1)

**Review of "The use of ground-based GNSS for atmospheric water vapour variation study in Papua New Guinea and its response to ENSO events."**

Response to the reviewer

Dear reviewer,

Thank you very much for your time in reviewing this manuscript. Your analysis and in-depth critical comments, suggestions and recommendations on this paper are well acknowledged and very much appreciated.

- Paragraph 3: The comments in the third and fourth paragraphs regarding the outdated literature review is well noted. It would be more appropriate for the manuscript to add special issues and articles submitted to ACP with a more updated GNSS PWV introduction as per reference tips and suggestions by the reviewer, and should elaborate more on newly evolved NWP models. Adding on, yes indeed the climatic description of the study area is referenced to old references as there is literally a huge void concerning meteorological and weather studies around the Papua New Guinean region in the 21$^{st}$ century, which is also one of the aims of this paper: by attempting to provide some information about the climatic change over this region through atmospheric water vapour studies.

- Paragraph 4: The reviewer is correct. Due to unfortunate circumstances during the start of this study, the authors we were forced to used ERA-Interim. However, the advantageous aspect of ERA5 that the reviewer raised on investigation the diurnal variability of PWV is noted and should be looked into. Therefore, this manuscript will be revised to apply ERA5.

- Paragraph 6: The suggestions and expected interpretation of presented results of radiosonde measurements is well acknowledged. A more careful investigation into the radiosonde types used at Momote station during the active period and moist bias shown after 2006 onwards will be investigated again with proper discussion. Also, the nil bias between GNSS and radiosonde will have to be elaborated too.

- Paragraph 7: Section 3.1, the Tm error analysis was included in the paper to validate the reliability of using Tm derived from ERA Interim by comparing with radiosonde at one of the locations, however it did not quantify the impact it has on PWV. Therefore, this section will to be removed.

  On that note, the three GPS stations are not collocated with meteorological sensors or neighbouring synoptic stations, and therefore reanalysis products of Tm and Ps from ERA-Interim were used which provide the optimal data source for PWV retrieval. These two PWV datasets are considered independent as GNSS data are not assimilated into ERA-Interim, allowing for an independent validation of ERA- Interim. But further discussion on this will have to be elaborated better in the revised manuscript using ERA5 PWV, which will expect correlate to some extent.

- Paragraph 8: Concern and critic for short PWV trends is well noted. PWV trends derived from radiosondes between 7 to 13 years are not reliable nor do they represent PWV variation from a climatic standpoint properly due to gaps in radiosonde PWV data and the large uncertainties in the trend. Although the GPS stations time span are rather short to the required 30 to 40 years to detect significant PWV trends as published by Alshawaf et al., the trend estimates in the research do demonstrate the potential for GNSS PWV to provide accurate information should the adequate length be achieved. The physical interpretation for GNSS and ERA- Interim PWV trends per year does also demonstrate the change in atmospheric moisture due to rising temperatures per year.

- Paragraph 9: This part of the paper attempted to study the behaviour of GNSS PWV during major ENSO activities, as presented in Fig. 6 and 7. With the predefined indicators confirming the occurrences of ENSO, the two NINO regions were selected according to their geographical locations to Papua New Guinea, and their differences in indicators in SST anomalies values - however, your point on the different phases and the significance of the correlations is well noted and will require proper discussion/ changes.

- Weekly OISSTa was seen as more suitable to account for missing GNSS data per week during ENSO event and because OISST is beneficial with the higher resolution in SST during the ENSO event. The discussion about precipitation to explain the PWV and ENSO was referenced to reports and newspaper articles during these events, as precipitation datasets from the local meteorological office was not reliable during these events. Should a precipitation dataset be taken from climate models, further assessments of the rainfall dataset would bring us off the research scope.

- Paragraph 10: The discussion and conclusion of the paper was intended to bring out the large research gap in GNSS Meteorology and extreme weather monitoring in general over this particular region of the world. However, suggestions and recommendations by the reviewer on reporting significant findings as well as the need for a major revision of the manuscript are well noted and appreciated, which the authors will now look into again.